# Color Constancy by Learning to Predict Chromaticity from Luminance

**Ayan Chakrabarti**
Toyota Technological Institute at Chicago
6045 S. Kenwood Ave., Chicago, IL 60637
ayanc@ttic.edu

## Abstract

Color constancy is the recovery of true surface color from observed color, and requires estimating the chromaticity of scene illumination to correct for the bias it induces. In this paper, we show that the per-pixel color statistics of natural scenes—without any spatial or semantic context—can by themselves be a powerful cue for color constancy. Specifically, we describe an illuminant estimation method that is built around a "classifier" for identifying the true chromaticity of a pixel given its luminance (absolute brightness across color channels). During inference, each pixel's observed color restricts its true chromaticity to those values that can be explained by one of a candidate set of illuminants, and applying the classifier over these values yields a distribution over the corresponding illuminants. A global estimate for the scene illuminant is computed through a simple aggregation of these distributions across all pixels. We begin by simply defining the luminance-to-chromaticity classifier by computing empirical histograms over discretized chromaticity and luminance values from a training set of natural images. These histograms reflect a preference for hues corresponding to smooth reflectance functions, and for achromatic colors in brighter pixels. Despite its simplicity, the resulting estimation algorithm outperforms current state-of-the-art color constancy methods. Next, we propose a method to learn the luminance-to-chromaticity classifier "end-to-end". Using stochastic gradient descent, we set chromaticity-luminance likelihoods to minimize errors in the final scene illuminant estimates on a training set. This leads to further improvements in accuracy, most significantly in the tail of the error distribution.

## 1  Introduction

The spectral distribution of light reflected off a surface is a function of an intrinsic material property of the surface—its reflectance—and also of the spectral distribution of the light illuminating the surface. Consequently, the observed color of the same surface under different illuminants in different images will be different. To be able to reliably use color computationally for identifying materials and objects, researchers are interested in deriving an encoding of color from an observed image that is invariant to changing illumination. This task is known as color constancy, and requires resolving the ambiguity between illuminant and surface colors in an observed image. Since both of these quantities are unknown, much of color constancy research is focused on identifying models and statistical properties of natural scenes that are informative for color constancy. While pschophysical experiments have demonstrated that the human visual system is remarkably successful at achieving color constancy [1], it remains a challenging task computationally.

Early color constancy algorithms were based on relatively simple models for pixel colors. For example, the gray world method [2] simply assumed that the average true intensities of different color channels across all pixels in an image would be equal, while the white-patch retinex method [3]

assumed that the true color of the brightest pixels in an image is white. Most modern color constancy methods, however, are based on more complex reasoning with higher-order image features. Many methods [4, 5, 6] use models for image derivatives instead of individual pixels. Others are based on recognizing and matching image segments to those in a training set to recover true color [7]. A recent method proposes the use of a multi-layer convolutional neural network (CNN) to regress from image patches to illuminant color. There are also many "combination-based" color constancy algorithms, that combine illuminant estimates from a number of simpler "unitary" algorithms [8, 9, 10, 11], sometimes using image features to give higher weight to the outputs of some subset of methods.

In this paper, we demonstrate that by appropriately modeling and reasoning with the statistics of individual pixel colors, one can computationally recover illuminant color with high accuracy. We consider individual pixels in isolation, where the color constancy task reduces to discriminating between the possible choices of true color for the pixel that are feasible given the observed color and a candidate set of illuminants. Central to our method is a function that gives us the relative likelihoods of these true colors, and therefore a distribution over the corresponding candidate illuminants. Our global estimate for the scene illuminant is then computed by simply aggregating these distributions across all pixels in the image.

We formulate the likelihood function as one that measures the conditional likelihood of true pixel chromaticity given observed luminance, in part to be agnostic to the scalar (i.e., color channel-independent) ambiguity in observed color intensities. Moreover, rather than committing to a parametric form, we quantize the space of possible chromaticity and luminance values, and define the function over this discrete domain. We begin by setting the conditional likelihoods purely empirically, based simply on the histograms of true color values over all pixels in all images across a training set. Even with this purely empirical approach, our estimation algorithm yields estimates with higher accuracy than current state-of-the-art methods. Then, we investigate learning the per-pixel belief function by optimizing an objective based on the accuracy of the final global illuminant estimate. We carry out this optimization using stochastic gradient descent, and using a sub-sampling approach (similar to "dropout" [12]) to improve generalization beyond the training set. This further improves estimation accuracy, without adding to the computational cost of inference.

## 2 Preliminaries

Assuming Lambertian reflection, the spectral distribution of light reflected by a material is a product of the distribution of the incident light and the material's reflectance function. The color intensity vector $\mathbf{v}(n) \in \mathbb{R}^3$ recorded by a tri-chromatic sensor at each pixel $n$ is then given by

$$\mathbf{v}(n) = \int \kappa(n, \lambda)\ell(n, \lambda)\, s(n)\, \mathbf{\Pi}(\lambda)\, d\lambda, \tag{1}$$

where $\kappa(n, \lambda)$ is the reflectance at $n$, $\ell(n, \lambda)$ is the spectral distribution of the incident illumination, $s(n)$ is a geometry-dependent shading factor, and $\mathbf{\Pi}(\lambda) \in \mathbb{R}^3$ denotes the spectral sensitivities of the color sensors. Color constancy is typically framed as the task of computing from $\mathbf{v}(n)$ the corresponding color intensities $\mathbf{x}(n) \in \mathbb{R}^3$ that would have been observed under some canonical illuminant $\ell_{\mathrm{ref}}$ (typically chosen to be $\ell_{\mathrm{ref}}(\lambda) = 1$). We will refer to $\mathbf{x}(n)$ as the "true color" at $n$.

Since (1) involves a projection of the full incident light spectrum on to the three filters $\mathbf{\Pi}(\lambda)$, it is not generally possible to recover $\mathbf{x}(n)$ from $\mathbf{v}(n)$ even with knowledge of the illuminant $\ell(n, \lambda)$. However, a commonly adopted approximation (shown to be reasonable under certain assumptions [13]) is to relate the true and observed colors $\mathbf{x}(n)$ and $\mathbf{v}(n)$ by a simple per-channel adaptation:

$$\mathbf{v}(n) = \mathbf{m}(n) \circ \mathbf{x}(n), \tag{2}$$

where $\circ$ refers to the element-wise Hadamard product, and $\mathbf{m}(n) \in \mathbb{R}^3$ depends on the illuminant $\ell(n, \lambda)$ (for $\ell_{\mathrm{ref}}$, $\mathbf{m} = [1, 1, 1]^T$). With some abuse of terminology, we will refer to $\mathbf{m}(n)$ as the illuminant in the remainder of the paper. Moreover, we will focus on the single-illuminant case in this paper, and assume $\mathbf{m}(n) = \mathbf{m}$, $\forall n$ in an image. Our goal during inference will be to estimate this global illuminant $\mathbf{m}$ from the observed image $\mathbf{v}(n)$. The true color image $\mathbf{x}(n)$ can then simply be recovered as $\mathbf{m}^{-1} \circ \mathbf{v}(n)$, where $\mathbf{m}^{-1} \in \mathbb{R}^3$ denotes the element-wise inverse of $\mathbf{m}$.

Note that color constancy algorithms seek to resolve the ambiguity between $\mathbf{m}$ and $\mathbf{x}(n)$ in (2) only up to a channel-independent scalar factor. This is because scalar ambiguities show up in $\mathbf{m}$ between

$\ell$ and $\ell_{\text{ref}}$ due to light attenuation, between $\mathbf{x}(n)$ and $\kappa(n)$ due to the shading factor $s(n)$, and in the observed image $\mathbf{v}(n)$ itself due to varying exposure settings. Therefore, the performance metric typically used is the angular error $\cos^{-1}\left(\frac{\mathbf{m}^T\bar{\mathbf{m}}}{\|\mathbf{m}\|_2\|\bar{\mathbf{m}}\|_2}\right)$ between the true and estimated illuminant vectors $\mathbf{m}$ and $\bar{\mathbf{m}}$.

**Database** For training and evaluation, we use the database of 568 natural indoor and outdoor images captured under various illuminants by Gehler et al. [14]. We use the version from Shi and Funt [15] that contains linear images (without gamma correction) generated from the RAW camera data. The database contains images captured with two different cameras (86 images with a Canon 1D, and 482 with a Canon 5D). Each image contains a color checker chart placed in the image, with its position manually labeled. The colors of the gray squares in the chart are taken to be the value of the true illuminant $m$ for each image, which can then be used to correct the image to get true colors at each pixel (of course, only up to scale). The chart is masked out during evaluation. We use k-fold cross-validation over this dataset in our experiments. Each fold contains images from both cameras corresponding to one of k roughly-equal partitions of each camera's image set (ordered by file name/order of capture). Estimates for images in each fold are based on training only with data from the remaining folds. We report results with three- and ten-fold cross-validation. These correspond to average training set sizes of 379 and 511 images respectively.

## 3   Color Constancy with Pixel-wise Chromaticity Statistics

A color vector $\mathbf{x} \in \mathbb{R}^3$ can be characterized in terms of (1) its *luminance* $\|\mathbf{x}\|_1$, or absolute brightness across color channels; and (2) its *chromaticity*, which is a measure of the relative ratios between intensities in different channels. While there are different ways of encoding chromaticity, we will do so in terms of the unit vector $\hat{\mathbf{x}} = \mathbf{x}/\|\mathbf{x}\|_2$ in the direction of $\mathbf{x}$. Note that since intensities can not be negative, $\hat{\mathbf{x}}$ is restricted to lie on the non-negative eighth of the unit sphere $\mathbb{S}_+^2$. Remember from Sec. 2 that our goal is to resolve the ambiguity between the true colors $\mathbf{x}(n)$ and the illuminant $\mathbf{m}$ only up to scale. In other words, we need only estimate the illuminant chromaticity $\hat{\mathbf{m}}$ and true chromaticities $\hat{\mathbf{x}}(n)$ from the observed image $\mathbf{v}(n)$, which we can relate from (2) as

$$\hat{\mathbf{x}}(n) = \frac{\mathbf{x}(n)}{\|\mathbf{x}(n)\|_2} = \frac{\hat{\mathbf{m}}^{-1} \circ \mathbf{v}(n)}{\|\hat{\mathbf{m}}^{-1} \circ \mathbf{v}(n)\|_2} \triangleq g(\mathbf{v}(n), \hat{\mathbf{m}}). \tag{3}$$

A key property of natural illuminant chromaticities is that they are known to take a fairly restricted set of values, close to a one-dimensional locus predicted by Planck's radiation law [16]. To be able to exploit this, we denote $\mathcal{M} = \{\hat{\mathbf{m}}_i\}_{i=1}^M$ as the set of possible values for illuminant chromaticity $\hat{\mathbf{m}}$, and construct it from a training set. Specifically, we quantize[1] the chromaticity vectors $\{\hat{\mathbf{m}}_t\}_{t=1}^T$ of the illuminants in the training set, and let $\mathcal{M}$ be the set of unique chromaticity values. Additionally, we define a "prior" $b_i = \log(n_i/T)$ over this candidate set, based on the number $n_i$ of training illuminants that were quantized to $\hat{\mathbf{m}}_i$.

Given the observed color $\mathbf{v}(n)$ at a single pixel $n$, the ambiguity in $\hat{\mathbf{m}}$ across the illuminant set $\mathcal{M}$ translates to a corresponding ambiguity in the true chromaticity $\hat{\mathbf{x}}(n)$ over the set $\{g(\mathbf{v}(n), \hat{\mathbf{m}}_i)\}_i$. Figure 1(a) illustrates this ambiguity for a few different observed colors $v$. We note that while there is significant angular deviation within the set of possible true chromaticity values for any observed color, values in each set lie close to a one dimensional locus in chromaticity space. This suggests that the illuminants in our training set are indeed a good fit to Planck's law[2].

The goal of our work is to investigate the extent to which we can resolve the above ambiguity in true chromaticity on a per-pixel basis, without having to reason about the pixel's spatial neighborhood or semantic context. Our approach is based on computing a likelihood distribution over the possible values of $\hat{\mathbf{x}}(n)$, given the observed luminance $\|\mathbf{v}(n)\|_1$. But as mentioned in Sec. 2, there is considerable ambiguity in the scale of observed color intensities. We address this partially by applying a simple per-image global normalization to the observed luminance to define

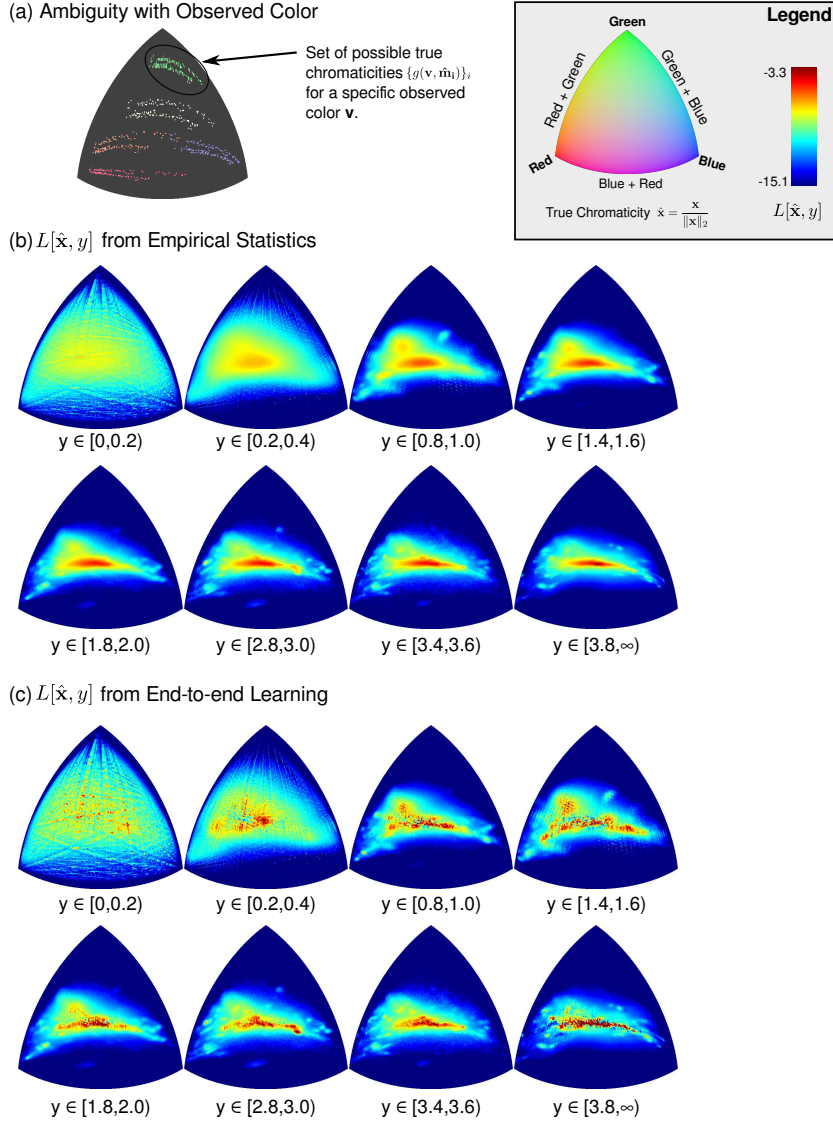

Figure 1: Color Constancy with Per-pixel Chromaticity-luminance distributions of natural scenes. (a) Ambiguity in true chromaticity given observed color: each set of points corresponds to the possible true chromaticity values (location in $\mathbb{S}^2_+$, see legend) consistent with the pixel's observed chromaticity (color of the points) and different candidate illuminants $\hat{\mathbf{m}}_i$. (b) Distributions over different values for true chromaticity of a pixel conditioned on its observed luminance, computed as empirical histograms over the training set. Values $y$ are normalized per-image by the median luminance value over all pixels. (c) Corresponding distributions learned with end-to-end training to maximize accuracy of overall illuminant estimation.

$y(n) = \|\mathbf{v}(n)\|_1 / \text{median}\{\|\mathbf{v}(n')\|_1\}_{n'}$. This very roughly compensates for variations across images due to exposure settings, illuminant brightness, etc. However, note that since the normalization is global, it does not compensate for variations due to shading.

The central component of our inference method is a function $L[\hat{\mathbf{x}}, y]$ that encodes the belief that a pixel with normalized observed luminance $y$ has true chromaticity $\hat{\mathbf{x}}$. This function is defined over a discrete domain by quantizing both chromaticity and luminance values: we clip luminance values $y$ to four (i.e., four times the median luminance of the image) and quantize them into twenty equal sized bins; and for chromaticity $\hat{\mathbf{x}}$, we use a much finger quantization with $2^{14}$ equal-sized bins in $\mathbb{S}^2_+$ (see supplementary material for details). In this section, we adopt a purely empirical approach

and define $L[\hat{\mathbf{x}}, y]$ as $L[\hat{\mathbf{x}}, y] = \log\left(N_{\hat{\mathbf{x}}, y} / \sum_{\hat{\mathbf{x}}'} N_{\hat{\mathbf{x}}', y}\right)$, where $N_{\hat{\mathbf{x}}, y}$ is the number of pixels across all pixels in a set of images in a training set that have true chromaticity $\hat{\mathbf{x}}$ and observed luminance $y$.

We visualize these empirical versions of $L[\hat{\mathbf{x}}, y]$ for a subset of the luminance quantization levels in Fig. 1(b). We find that in general, desaturated chromaticities with similar intensity values in all color channels are most common. This is consistent with findings of statistical analysis of natural spectra [17], which shows the "DC" component (flat across wavelength) to be the one with most variance. We also note that the concentration of the likelihood mass in these chromaticities increasing for higher values of luminance $y$. This phenomenon is also predicted by traditional intuitions in color science: materials are brightest when they reflect most of the incident light, which typically occurs when they have a flat reflectance function with all values of $\kappa(\lambda)$ close to one. Indeed, this is what forms the basis of the white-patch retinex method [3]. Amongst saturated colors, we find that hues which combine green with either red or blue occur more frequently than primary colors, with pure green and combinations of red and blue being the least common. This is consistent with findings that reflectance functions are usually smooth (PCA on pixel spectra in [17] revealed a Fourier-like basis). Both saturated green and red-blue combinations would require the reflectance to have either a sharp peak or crest, respectively, in the middle of the visible spectrum.

We now describe a method that exploits the belief function $L[\hat{\mathbf{x}}, y]$ for illuminant estimation. Given the observed color $\mathbf{v}(n)$ at a pixel $n$, we can obtain a distribution $\{L[g(\mathbf{v}(n), \hat{\mathbf{m}}_i), y(n)]\}_i$ over the set of possible true chromaticity values $\{g(\mathbf{v}(n), \hat{\mathbf{m}}_i)\}_i$, which can also be interpreted as a distribution over the corresponding illuminants $\hat{\mathbf{m}}_i$. We then simply aggregate these distributions across all pixels $n$ in the image, and define the global probability of $\hat{\mathbf{m}}_i$ being the scene illuminant $\mathbf{m}$ as $p_i = \exp(l_i) / \left(\sum_{i'} \exp(l_{i'})\right)$, where

$$l_i = \frac{\alpha}{N} \sum_n L[g(\mathbf{v}(n), \hat{\mathbf{m}}_i), y(n)] + \beta b_i, \tag{4}$$

$N$ is the total number of pixels in the image, and $\alpha$ and $\beta$ are scalar parameters. The final illuminant chromaticity estimate $\bar{\mathbf{m}}$ is then computed as

$$\bar{\mathbf{m}} = \underset{\mathbf{m}', \|\mathbf{m}'\|_2 = 1}{\arg\min} \mathbb{E}\left[\cos^{-1}(\mathbf{m}^T \mathbf{m}')\right] \approx \underset{\mathbf{m}', \|\mathbf{m}'\|_2 = 1}{\arg\max} \mathbb{E}[\mathbf{m}^T \mathbf{m}'] = \frac{\sum_i p_i \mathbf{m}_i}{\|\sum_i p_i \mathbf{m}_i\|_2}. \tag{5}$$

Note that (4) also incorporates the prior $b_i$ over illuminants. We set the parameters $\alpha$ and $\beta$ using a grid search, to values that minimize mean illuminant estimation error over the training set. The primary computational cost of inference is in computing the values of $\{l_i\}$. We pre-compute values of $g(\hat{\mathbf{x}}, \hat{\mathbf{m}})$ using (3) over the discrete domain of quantized chromaticity values for $\hat{\mathbf{x}}$ and the candidate illuminant set $\mathcal{M}$ for $\hat{\mathbf{m}}$. Therefore, computing each $l_i$ essentially only requires the addition of $N$ numbers from a look-up table. We need to do this for all $M = |\mathcal{M}|$ illuminants, where summations for different illuminants can be carried out in parallel. Our implementation takes roughly 0.3 seconds for a 9 mega-pixel image, on a modern Intel 3.3GHz CPU with 6 cores, and is available at http://www.ttic.edu/chakrabarti/chromcc/.

This empirical version of our approach bears some similarity to the Bayesian method of [14] that is based on priors for illuminants, and for the likelihood of different true reflectance values being present in a scene. However, the key difference is our modeling of true chromaticity *conditioned* on luminance that explicitly makes estimation agnostic to the absolute scale of intensity values. We also reason with all pixels, rather than the set of unique colors in the image.

**Experimental Results.** Table 1 compares the performance of illuminant estimation with our method (see rows labeled "Empirical") to the current state-of-the-art, using different quantiles of angular error across the Gehler-Shi database [14, 15]. Results for other methods are from the survey by Li et al. [18]. (See the supplementary material for comparisons to some other recent methods).

We show results with both three- and ten-fold cross-validation. We find that our errors with three-fold cross-validation have lower mean, median, and tri-mean values than those of the best performing state-of-the-art method from [8], which combines illuminant estimates from twelve different "unitary" color-constancy method (many of which are also listed in Table 1) using support-vector regression. The improvement in error is larger with respect to the other combination methods [8, 9, 10, 11], as well as those based the statistics of image derivatives [4, 5, 6]. Moreover, since our method has more parameters than most previous algorithms ($L[\hat{\mathbf{x}}, y]$ has $2^{14} \times 20 \approx 300\text{k}$ entries), it is likely

Table 1: Quantiles of Angular Error for Different Methods on the Gehler-Shi Database [14, 15]

| Method | Mean | Median | Tri-mean | 25%-ile | 75%-ile | 90%-ile |
|---:|:---:|:---:|:---:|:---:|:---:|:---:|
| Bayesian [14] | 6.74° | 5.14° | 5.54° | 2.42° | 9.47° | 14.71° |
| Gamut Mapping [20] | 6.00° | 3.98° | 4.52° | 1.71° | 8.42° | 14.74° |
| Deriv. Gamut Mapping [4] | 5.96° | 3.83° | 4.32° | 1.68° | 7.95° | 14.72° |
| Gray World [2] | 4.77° | 3.63° | 3.92° | 1.81° | 6.63° | 10.59° |
| Gray Edge$^{(1,1,6)}$ [5] | 4.19° | 3.28° | 3.54° | 1.87° | 5.72° | 8.60° |
| SV-Regression [21] | 4.14° | 3.23° | 3.35° | 1.68° | 5.27° | 8.87° |
| Spatio-Spectral [6] | 3.99° | 3.24° | 3.45° | 2.38° | 4.97° | 7.50° |
| Scene Geom. Comb. [9] | 4.56° | 3.15° | 3.46° | 1.41° | 6.12° | 10.39° |
| Nearest-30% Comb. [10] | 4.26° | 2.95° | 3.19° | 1.49° | 5.39° | 9.67° |
| Classifier-based Comb. [11] | 3.83° | 2.75° | 2.93° | 1.34° | 4.89° | 8.19° |
| Neural Comb. (ELM) [8] | 3.43° | 2.37° | 2.62° | 1.21° | 4.53° | 6.97° |
| SVR-based Comb. [8] | 2.98° | 1.97° | 2.35° | 1.13° | 4.33° | 6.37° |

**Proposed**

| | Mean | Median | Tri-mean | 25%-ile | 75%-ile | 90%-ile |
|---:|:---:|:---:|:---:|:---:|:---:|:---:|
| (3-Fold)  Empirical | 2.89° | 1.89° | 2.15° | 1.15° | 3.68° | 6.24° |
| End-to-end Trained | 2.56° | 1.67° | 1.89° | 0.91° | 3.30° | 5.56° |
| (10-Fold)  Empirical | 2.55° | 1.58° | 1.83° | 0.85° | 3.30° | 5.74° |
| End-to-end Trained | 2.20° | 1.37° | 1.53° | 0.69° | 2.68° | 4.89° |

to benefit from more training data. We find this to indeed be the case, and observe a considerable decrease in error quantiles when we switch to ten-fold cross-validation.

Figure. 2 shows estimation results with our method for a few sample images. For each image, we show the input image (indicating the ground truth color chart being masked out) and the output image with colors corrected by the global illuminant estimate. To visualize the quality of contributions from individual pixels, we also show a map of angular errors for illuminant estimates from individual pixels. These estimates are based on values of $l_i$ computed by restricting the summation in (4) to individual pixels. We find that even these pixel-wise estimates are fairly accurate for a lot of pixels, even when it's true color is saturated (see cart in first row). Also, to evaluate the weight of these per-pixel distributions to the global $l_i$, we show a map of their variance on a per-pixel basis. As expected from Fig. 1(b), we note higher variances in relatively brighter pixels. The image in the last row represents one of the poorest estimates across the entire dataset (higher than $90\%-$ile). Note that much of the image is in shadow, and contain only a few distinct (and likely atypical) materials.

## 4   Learning $L[\hat{\mathbf{x}}, y]$ End-to-end

While the empirical approach in the previous section would be optimal if pixel chromaticities in a typical image were infact i.i.d., that is clearly not the case. Therefore, in this section we propose an alternate approach method to setting the beliefs in $L[\hat{\mathbf{x}}, y]$, that optimizes for the accuracy of the final global illuminant estimate. However, unlike previous color constancy methods that explicitly model statistical co-dependencies between pixels—for example, by modeling spatial derivatives [4, 5, 6], or learning functions on whole-image histograms [21]—we retain the overall parametric "form" by which we compute the illuminant in (4). Therefore, even though $L[\hat{\mathbf{x}}, y]$ itself is learned through knowledge of co-occurence of chromaticities in natural images, estimation of the illuminant during inference is still achieved through a simple aggregation of per-pixel distributions.

Specifically, we set the entries of $L[\hat{\mathbf{x}}, y]$ to minimize a cost function $C$ over a set of training images:

$$C(L) = \sum_{t=1}^{T} C^t(L), \qquad C^t = \sum_i \cos^{-1}(\hat{\mathbf{m}}_i^T \hat{\mathbf{m}}^t)\, p_i^t, \qquad (6)$$

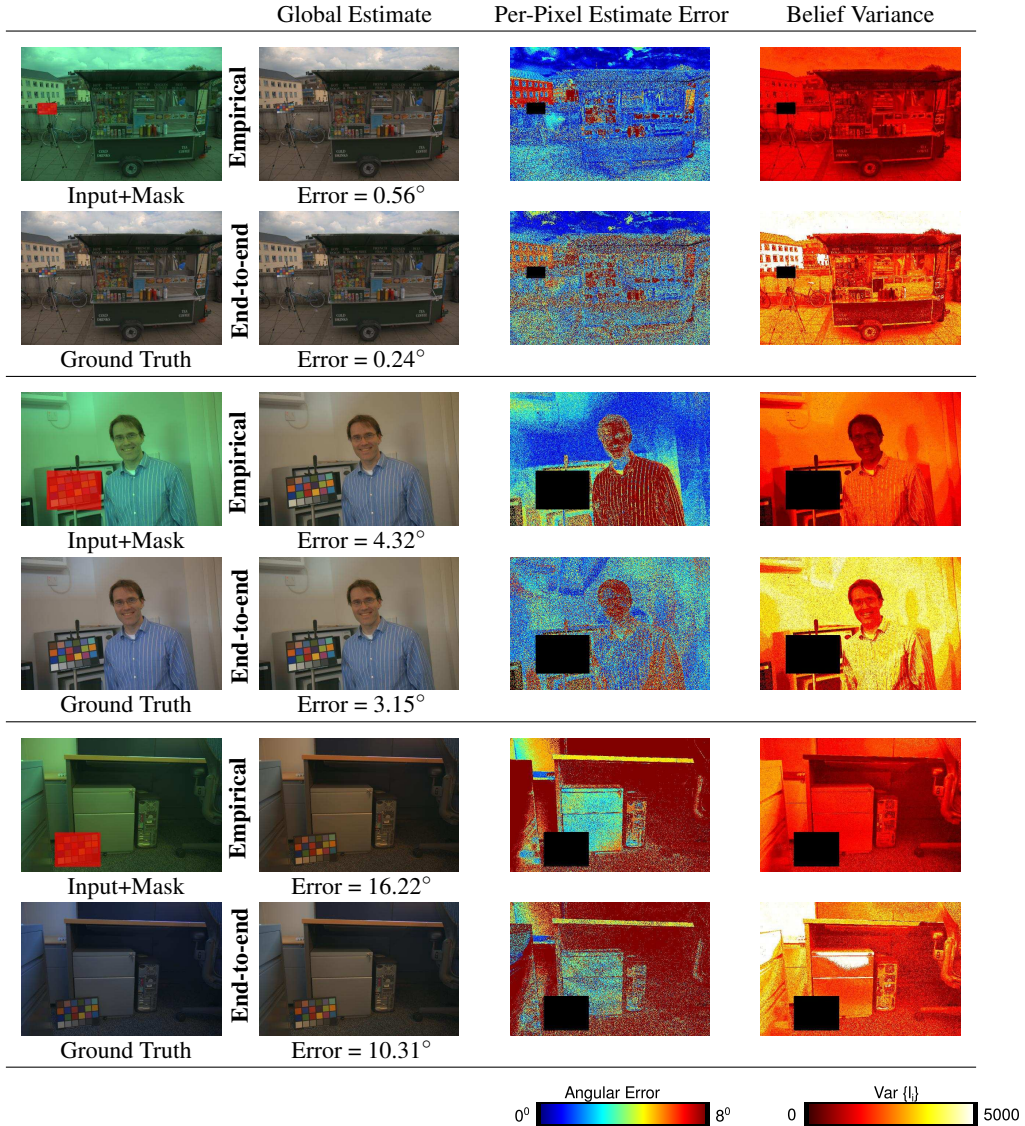

Figure 2: Estimation Results on Sample Images. Along with output images corrected with the global illuminant estimate from our methods, we also visualize illuminant information extracted at a local level. We show a map of the angular error of pixel-wise illuminant estimates (i.e., computed with $l_i$ based on distributions from only a single pixel). We also show a map of the variance $\mathrm{Var}(\{l_i\}_i)$ of these beliefs, to gauge the weight of their contributions to the global illuminant estimate.

where $\hat{\mathbf{m}}^t$ is the true illuminant chromaticity of the $t^{th}$ training image, and $p_i^t$ is computed from the observed colors $v^t(n)$ using (4). We augment the training data available to us by "re-lighting" each image with different illuminants from the training set. We use the original image set and six re-lit copies for training, and use a seventh copy for validation.

We use stochastic gradient descent to minimize (6). We initialize $L$ to empirical values as described in the previous section (for convenience, we multiply the empirical values by $\alpha$, and then set $\alpha = 1$ for computing $l_i$), and then consider individual images from the training set at each iteration. We make multiple passes through the training set, and at each iteration, we randomly sub-sample the pixels from each training image. Specifically, we only retain $1/128$ of the total pixels in the image by randomly sub-sampling $16 \times 16$ patches at a time. This approach, which can be interpreted as being similar to "dropout" [12], prevents over-fitting and improves generalization.

Derivatives of the cost function $C^t$ with respect to the current values of beliefs $L[\hat{\mathbf{x}}, y]$ are given by

$$\frac{\partial C^t}{\partial L[\hat{\mathbf{x}}, y]} = \frac{1}{N} \sum_i \left( \sum_n \delta\left(g(v^t(n), \hat{\mathbf{m}}_i) = \hat{\mathbf{x}}\right) \delta\left(y^t(n) = y\right) \right) \times \frac{\partial C^t}{\partial l_i^t}, \tag{7}$$

where $\quad \dfrac{\partial C^t}{\partial l_i^t} = p_i^t \left( \cos^{-1}(\hat{\mathbf{m}}_i^T \hat{\mathbf{m}}^t) - C^t \right).$ $\tag{8}$

We use momentum to update the values of $L[\hat{\mathbf{x}}, y]$ at each iteration based on these derivative as

$$L[\hat{\mathbf{x}}, y] = L[\hat{\mathbf{x}}, y] - L^\nabla[\hat{\mathbf{x}}, y], \quad L^\nabla[\hat{\mathbf{x}}, y] = r \frac{\partial C^t}{\partial L[\hat{\mathbf{x}}, y]} + \mu L_*^\nabla[\hat{\mathbf{x}}, y], \tag{9}$$

where $L_*^\nabla[\hat{\mathbf{x}}, y]$ is the previous update value, $r$ is the learning rate, and $\mu$ is the momentum factor. In our experiments, we set $\mu = 0.9$, run stochastic gradient descent for 20 epochs with $r = 100$, and another 10 epochs with $r = 10$. We retain the values of $L$ from each epoch, and our final output is the version that yields the lowest mean illuminant estimation error on the validation set.

We show the belief values learned in this manner in Fig. 1(c). Notice that although they retain the overall biases towards desaturated colors and combined green-red and green-blue hues, they are less "smooth" than their empirical counterparts in Fig. 1(b)—in many instances, there are sharp changes in the values $L[\hat{\mathbf{x}}, y]$ for small changes in chromaticity. While harder to interpret, we hypothesize that these variations result from shifting beliefs of specific $(\hat{\mathbf{x}}, y)$ pairs to their neighbors, when they correspond to incorrect choices within the ambiguous set of specific observed colors.

**Experimental Results.** We also report errors when using these end-to-end trained versions of the belief function $L$ in Table 1, and find that they lead to an appreciable reduction in error in comparison to their empirical counterparts. Indeed, the errors with end-to-end training using three-fold cross-validation begin to approach those of the empirical version with ten-fold cross-validation, which has access to much more training data. Also note that the most significant improvements (for both three- and ten-fold cross-validation) are in "outlier" performance, i.e., in the 75 and 90%-ile error values. Color constancy methods perform worst on images that are dominated by a small number of materials with ambiguous chromaticity, and our results indicate that end-to-end training increases the reliability of our estimation method in these cases.

We also include results for the end-to-end case for the example images in Figure. 2. For all three images, there is an improvement in the global estimation error. More interestingly, we see that the per-pixel error and variance maps now have more high-frequency variation, since $L$ now reacts more sharply to slight chromaticity changes from pixel to pixel. Moreover, we see that a larger fraction of pixels generate fairly accurate estimates by themselves (blue shirt in row 2). There is also a higher disparity in belief variance, including within regions that visually look homogeneous in the input, indicating that the global estimate is now more heavily influenced by a smaller fraction of pixels.

## 5   Conclusion and Future Work

In this paper, we introduced a new color constancy method that is based on a conditional likelihood function for the true chromaticity of a pixel, given its luminance. We proposed two approaches to learning this function. The first was based purely on empirical pixel statistics, while the second was based on maximizing accuracy of the final illuminant estimate. Both versions were found to outperform state-of-the-art color constancy methods, including those that employed more complex features and semantic reasoning. While we assumed a single global illuminant in this paper, the underlying per-pixel reasoning can likely be extended to the multiple-illuminant case, especially since, as we saw in Fig. 2, our method was often able to extract reasonable illuminant estimates from individual pixels. Another useful direction for future research is to investigate the benefits of using likelihood functions that are conditioned on *lightness*—estimated using an intrinsic image decomposition method—instead of normalized luminance. This would factor out the spatially-varying scalar ambiguity caused by shading, which could lead to more informative distributions.

### Acknowledgments

We thank the authors of [18] for providing estimation results of other methods for comparison. The author was supported by a gift from Adobe.

## Footnotes

[1] Quantization is over uniformly sized bins in $\mathbb{S}_+^2$. See supplementary material for details.

[2] In fact, the chromaticities appear to lie on two curves, that are slightly separated from each other. This separation is likely due to differences in the sensor responses of the two cameras in the Gehler-Shi dataset.

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
