[Supplementary Material]

**Supplementary Material**

**A: Quantizing the space of valid chromaticity values**

We adopt a standard approach to uniformly quantize unit vectors. Consider a chromaticity unit vector $\hat{\mathbf{x}} = [\hat{\mathbf{x}}_R, \hat{\mathbf{x}}_G, \hat{\mathbf{x}}_B]^T$. We parametrize this vector in terms of $u = \hat{\mathbf{x}}_G$, and $\theta = \tan^{-1}(\hat{\mathbf{x}}_B/\hat{\mathbf{x}}_R)$. Since the elements of $\hat{\mathbf{x}}$ are constrained to be positive, it follows that $u \in [0, 1]$ and $\theta \in [0, \pi/2]$. We uniformly quantize $u$ and $\theta$ in their respective domains, using $64$ bins for each when quantizing illuminant chromaticities to construct $\mathcal{M}$, and $128$ bins for true pixel chromaticities to define $L[\hat{\mathbf{x}}, y]$.

**B: Black-level offset issues with results of Exemplar-based [7] and deep-CNN methods [19]**

In this section, we compare our method to two recent semantic reasoning-based methods—exemplar-based [7] and deep-CNN [19]. While the illuminant estimates computed by these methods on the Gehler-Shi database were made available by their authors, unfortunately, they are based on training and testing with older incorrect versions of the ground-truth and intensity data for images from one of the cameras.

Specifically, the intensities in the image files from the Canon 5D camera in the Gehler-Shi dataset includes a "black-level" offset of 129, that needs to be subtracted from the intensities of all pixels in all color channels. This offset affects both the observed image data, as well as the ground truth computed from the color checker chart. While this was eventually clarified by the authors of [15] (and the ground-truth made available by them reflects this correction), there remain older versions of the ground truth without this correction, and the estimated illuminants for [7] and [19] made available are with respect to these old versions.

We attempt to compare the performance of our method with that of [7, 19] in two ways. First, we generate corrected estimates $\bar{\mathbf{m}}^*$ from the illuminant chromaticities $\bar{\mathbf{m}}$ estimated by these methods, by "subtracting" the effect of the black level offset. This is done based on the true "un-normalized" ground truth illuminants $m$ (i.e., the color of the gray squares in the color chart) as:

$$\bar{\mathbf{m}}^- = \frac{\bar{\mathbf{m}}}{|\bar{\mathbf{m}}|_1} \times (|\mathbf{m}|_1 + 129 \times 3) - [129, 129, 129]^T. \tag{10}$$

Essentially, we rendered the color of the gray square by multiplying the luminance of non-corrected ground-truth with the estimated illuminant $\bar{\mathbf{m}}$, and then subtracted the black level offset. We show the errors of these corrected illuminant estimates with respect to the actual ground truth in Table 2, and also copy the results of our method from Table 1.

Next, we do the reverse. We "add" the offset back to the illuminant estimates from our method:

$$\bar{\mathbf{m}}^+ = \frac{\bar{\mathbf{m}}}{|\bar{\mathbf{m}}|_1} \times |\mathbf{m}|_1 + [129, 129, 129]^T, \tag{11}$$

and compare these to the non-corrected ground truth with respect to which the results of [7] and [19] are reported. Table 3 provides this comparison.

We find that the proposed method outperforms [7] and [19] in both comparisons. However, the quantiles reported in both Table 2 and Table 3 for [7, 19] should be interpreted only as a conservative estimate of their performance. While the corrections in (10),(11) allowed us to report errors for these methods and ours with respect to a common ground truth, it does *not* correct for the fact that estimates for [7, 19] were computed on offset data. In particular, the presence of this offset means that the linear relationship (2) between observed colors and the scene illuminant no longer holds.

Table 2: Comparison with respect to actual ground-truth, with correction (10) applied to [7, 19].

| Method | Mean | Median | Tri-mean | 25%-ile | 75%-ile | 90%-ile |
|---|---|---|---|---|---|---|
| Exemplar-Based [7] | 3.66° | 2.91° | 3.04° | 1.52° | 4.81° | 7.22° |
| Deep-CNN [19] | 3.45° | 2.45° | 2.65° | 1.40° | 4.29° | 7.23° |

**Proposed**

| | Mean | Median | Tri-mean | 25%-ile | 75%-ile | 90%-ile |
|---|---|---|---|---|---|---|
| (3-Fold)   Empirical | 2.89° | 1.89° | 2.15° | 1.15° | 3.68° | 6.24° |
| End-to-end Trained | 2.56° | 1.67° | 1.89° | 0.91° | 3.30° | 5.56° |
| (10-Fold)   Empirical | 2.55° | 1.58° | 1.83° | 0.85° | 3.30° | 5.74° |
| End-to-end Trained | 2.20° | 1.37° | 1.53° | 0.69° | 2.68° | 4.89° |

Table 3: Comparison with respect to non-corrected ground-truth used in [7, 19], with correction (11) applied to our estimates.

| Method | Mean | Median | Tri-mean | 25%-ile | 75%-ile | 90%-ile |
|---|---|---|---|---|---|---|
| Exemplar-Based [7] | 2.89° | 2.27° | 2.42° | 1.29° | 3.84° | 5.68° |
| Deep-CNN [19] | 2.63° | 1.98° | 2.13° | 1.16° | 3.41° | 5.46° |

**Proposed**

| | Mean | Median | Tri-mean | 25%-ile | 75%-ile | 90%-ile |
|---|---|---|---|---|---|---|
| (3-Fold)   Empirical | 2.46° | 1.55° | 1.75° | 0.89° | 3.04° | 5.05° |
| End-to-end Trained | 2.21° | 1.39° | 1.54° | 0.71° | 2.65° | 4.79° |
| (10-Fold)   Empirical | 2.18° | 1.23° | 1.46° | 0.65° | 2.73° | 4.63° |
| End-to-end Trained | 1.91° | 1.11° | 1.24° | 0.54° | 2.21° | 4.08° |