[Reviews · NeurIPS 2015]

Submitted by Assigned_Reviewer_1

Quality:

This is a very nice paper - an elegant approach to a long studied problem. The methodology is clear, its motivations are well founded and experiments are well performed. There is a good amount of analysis (right most columns of Figure 2 are especially interesting) but I would want a bit more analysis for the relationship between the variance of estimators and the error at specific locations. Though I would expect these to be highly correlated, it does not seem that way in the figures.

Clarity:

The paper is very clearly written, easy to follow and uses notation well. Minor comments about this - I would use bold face characters for all properties which are vectors (such as chromaticity, pixels etc.). This would make the distinction between scalars and vectors easier. Additionally, some important equations are inline - line 158, 216 and line 237 certainly deserve their own numbered equation (or at least a clearer separation). In line 308, "it's" should be "its".

Originality:

An original approach to an old problem.

Significance:

An old and well studied problem which is interesting for a large part of the vision community. I'm not sure NIPS is an ideal audience for this work, but in terms of quality, it is certainly suitable.
Summary: This paper proposes a method for color constancy in image by learning about the conditional chromaticity distribution, conditioned on pixel luminance. This is done by simply modeling the empirical histograms in a training set, as well as globally optimizing the histograms over the training set using gradient descent on a cost function. Surprisingly enough, the luminance allows for informative predictions for the illumination chromaticity (up to the usual scaling constants) even when pixels are treated as independent. The method is quite fast to compute and results are pleasing as well as numerically impressive.

Submitted by Assigned_Reviewer_2

This is a well conducted study, with impressive results, summarized in an overall well written paper.

[... Part about readability of the paper; Not relevant anymore; Edited out ...]

I hope that the authors make their code available online should the paper be accepted.

Summary: The authors describe an illuminant estimation method based on identifying the true cromaticity of a pixel from its luminance using the fact that natural illuminant chromaticities only take on a restricted set of values because of Planck's radiation law. They estimate a luminance-to-chromaticity classifier from a training set of natural images and use that in inference to obtain a distribution over illuminants that are consistent with the observed pixels. A global estimate of the scene illuminant is computed by aggregating the distributions over illuminants over all pixels.

Submitted by Assigned_Reviewer_3

The algorithm presented here is simple and interesting. Pixel luminance, chrominance, and illumination chrominance are all histogrammed, and then evaluation is simply each pixel's luminance voting on each pixel's true chrominance for each of the "memorized" illuminations. The model can be trained generative by simply counting pixels in the training set, or can be trained end-to-end for a slight performance boost. This algorithm's simplicity and speed are appealing, and additionally it seems like it may be a useful building block for a more sophisticated spatially-varying illumination model.

I found the paper reasonable easy to read, though much of the math could use a revision to make things more clear and hopefully reduce the (unusually large) number of symbols which are defined for what seems to be a fairly simple algorithm. For example, why define x-hat and g()? is v(n) == v_n? Is { g(v(n), m-hat_i}_i really the easiest way to refer to that set --- and is that math even correct, given that i is used as a subscript twice? Perhaps some figures which correspond to the math would be useful, as the only variable which appears in a figure is L[].

The evaluation shows that this technique produces state-of-the-art results on the standard color checker dataset. The evaluation contains one component which I take issue with. When comparing the authors results to the results published in [19], we see that the authors very slightly outperform [19]. However, in this paper the authors have presented a modified set of error metrics, which in the supplement they explain are derived from the metrics in [7,19] subject to a correction, with the claim that this correction undoes an unfair advantage that the authors of [7,19] have achieved by using incorrectly black-leveled images. It may be true that [7,19] are evaluated incorrectly, but such an allegation should be taken very seriously, and should not be dealt with in this way. This number (as I understand it) assumes that the models of [7,19] were trained on the incorrect dataset and then evaluated on the correct dataset, which will obviously increase error metrics. But this does not mean that these numbers are indicative of how [7,19] would perform if trained *and* tested on the correct dataset. Indeed, given that incorrect black-level correction can completely ruin a color constancy algorithm, I would not be surprised in [7,19] actually performs better if trained and tested correctly. So I think the correction the authors present here is not valid.

Though it is unfortunate that [7,19] contains incorrect numbers which may be cited, we should not combat such innacuracies with other innacuracies. If published, the number presented here may be taken as truthful by future work, when in fact it is even more speculative and unreliable as the numbers presented in [7,19]. So I think the authors should not present their modified error numbers. The error metrics presented here should be the same as in [7,19], but perhaps indicated with an asterix or with a grayed-out font color that they are likely not trustworthy, with an explanation of why in the supplement. Then the onus will be on [7,19] to revise their results, and we wont have two contradictory (and equally incorrect) numbers for the performance of [7,19] circulating in publications. I don't suggest this to diminish the author's work, as even if [7,19] are taken on face value the work presented here outperforms it. This is simply a matter of scientific honesty and ettiquette that we should handle as carefully as possible.
Summary: The authors present an interesting color constancy algorithm which appears to outperform the state of the art on the standard color constancy dataset by a reasonable margin. The paper could be more clear and the evaluation has some issues, but otherwise the paper seems above-board.

Submitted by Assigned_Reviewer_4

The distribution of light reflected from a surface in a scene a function of the geometry, scene illuminance and the surface reflectance (true chromaticity). The problem of color constancy, is to estimate the surface reflectance from the observed scene, typically without being given either the geometry or scene illuminance information. This is of interest as a task that humans are relatively good at, and of importance for artificial vision systems to recognize and identify materials under variable lighting conditions.

The authors introduce a method for estimating the scene illuminance of an image (which then allows recovering the true chromaticity of all pixels). The method assumes a single, global luminance in the scene and uses a function L[x, y] encoding the belief that a pixel with an observed reflectance y has a true chromaticity x. The method also makes use of a prior on scene illuminances and an assumption of a single illuminance for the whole scene. Using a dataset containing calibrated images, they can learn L either by treating each pixel independently, or by directly minimizing the error in illuminance estimation on their training set with gradient descent. They show that their approach is generally superior to existing color constancy methods.

The manuscript is well-written and the authors compare their approach with a number of existing approaches. The method presented is pleasingly simple, makes reasonable assumptions and performs well. It is notable that this method makes no use of spatial structure in the scene and yet is able to outperform alternative approaches which do.

The approach used bears some similarities to the Bayesian estimation method used in reference 14 (the source of the image database). Although the author's compare against [14] in table 1, it would be helpful to include [14] in the discussion of existing methods on page 2.

For training of the end-to-end model the dataset is augmented by "re-lighting" each image with different illuminants to create 6 training images and a 7th validation image. It is unclear which hyper-parameters were chosen using this validation set. Importantly, I presume the images used for testing accuracy (table 1) are distinct (not just relit) from any used in training or validation (if not, this would be a serious flaw). This should be clarified briefly in the manuscript.

A weakness of this manuscript is that it is primarily of interest only for the machine vision community. The approach used is (from a machine learning point of view) fairly straightforward and the primary contribution is specific to the problem of color constancy.
Summary: End-to-end approach to the color constancy problem. Mostly of interest to the vision community.

Author Feedback
Author rebuttal: We thank the reviewers for their comments, and generally positive evaluation. While providing a fast and effective solution to a computer vision problem, we believe our paper fundamentally reveals a quality of natural image statistics. Our results indicate that despite the seeming diversity of natural colors, even their point-wise statistics---without spatial context or semantic reasoning---often carry enough information to disambiguate true color from the variations caused by illumination. This is a surprising fact, one that we believe will be of interest to many at NIPS.

Responses to questions/concerns in the reviews follow.

* Presentation: We appreciate the reviewers' suggestions for improving clarity, and will edit the paper to incorporate these. We will consolidate and simplify the notation, clarify the confusing terms reviewers pointed out (illuminant chromaticity is the chromaticity of the illuminant color vector), provide annotations for some of the mathematical terms in Fig 1, bold-face vectors, fix typos (\kappa, v_n should've been v(n)), etc.

* [7,19] Comparisons: We share R2's concerns and had debated various options on how to report [7,19]'s results.

To clarify:
--The black-level offset affected not only [7,19]'s input, but *also* their illuminant ground truth (GT). The errors reported in [7,19] are against the older incorrect GT (prior to the clarification by the authors of the dataset).
--[7,19]'s actual illuminant estimates (not just their errors) are available.
--Our goal was to report errors in Table 1 against the same (and correct) GT for all methods.
--Naturally, [7,19]'s estimates had very high errors with respect to the correct GT, as the methods were trained with offset data.
--The correction we applied (basically) subtracted this offset from their estimates before comparing to the correct GT, yielding comparatively lower errors that we reported in Table 1.
--We also did the "reverse" in Table 2: added the offset to our estimates and compared to the older (incorrect) GT.

But R2 is right: no correction can compensate for [7,19] being trained on incorrect data. But we also hesitate to directly include their original errors (their est. vs their incorrect GT) in Table 1, since it also includes other methods. As R2 notes, out-performing [7,19] isn't critical to our own narrative, but we don't want to imply that [7,19] are better than, say, the two methods from [8].

We feel the cleanest and most up-front solution is to remove [7,19] completely from Table 1, annotating it to say that comparisons to [7,19] are reported in supp. section. We will explain the offset issue in the supp sec., and provide all the different metrics for [7,19] vs our method in Table 2---[7,19]'s original numbers against their own GT, their corrected estimates against the true GT, as well as our errors against their GT. This should provide readers with enough information to draw their own conclusions, while avoiding misinterpretation since all numbers are in the same table.

* Others:

R3: We'll add discussion of [14] (the crucial difference is our chromaticity-based representation)

R3: The validation set was used to check for overfitting on the training set (line 395). The re-lit copies for training and validation were all generated from the original training set, and did not use *any* image or illuminant data from the test set.

R5: Code will be made available on publication.

R6: We feel there may have been a misunderstanding. Evaluation is based solely on the accuracy of predicting the scene illuminant, i.e., the transform parameters determined from the masked-out ground-truth, and not on the accuracy of predicting true colors ("data labels" for remaining pixels in the image).

The assumption that this transform is common to all pixels is used internally during training (to generate the true colors / "data labels" for all pixels), and during inference, when estimating this illuminant from the remaining image. Violations of the assumption would therefore hinder evaluation performance, rather than provide an advantage. This assumption is also made by all other methods in Table 1, so we'd say the comparisons are fair. Note that this is the standard evaluation methodology for the benchmark.